# Co-Creation of a Center for a Regenerative Future

**Sarah M. Bexell** [1], **Dean Saitta** [2], **Anna Sher** [3] **and Paul Sutton** [4,*]

1 Graduate School of Social Work, University of Denver, Denver, CO 80208, USA; sarah.bexell@du.edu
2 Department of Anthropology, University of Denver, Denver, CO 80208, USA; dean.saitta@du.edu
3 Department of Biological Sciences, University of Denver, Denver, CO 80208, USA; anna.sher@du.edu
4 Department of Geography, University of Denver, Denver, CO 80208, USA
* Correspondence: paul.sutton@du.edu

**Abstract:** We present the ideas, conditions, and environments that motivated our co-creation of a Center for a Regenerative Future at the University of Denver. There is an emerging consensus among scholars and a widening realization among younger generations that the concept of sustainability has exhausted its utility as a framework and rhetorical narrative for creating a viable future for humanity. Growing levels of eco-anxiety related to climate change, loss of biodiversity, and their social and economic consequences suggest that efforts to achieve 'sustainability' or 'sustainable development' are not succeeding. Dominant sustainability paradigms typically rest on an anthropocentric culture–nature dualism and a mechanistic worldview that perpetuates a growth-based economic system that is socially inequitable and ecologically destructive. Regenerative paradigms offer holistic understandings of Earth systems, with accompanying commitments to social and ecological justice. They support the development of resilient communities that allow for wider economic prosperity, security, and global environmental healing. We share the successes, challenges, and experiences associated with our effort to create a Center for a Regenerative Future in the hope that others can leverage this information to build their own regenerative institutions, communities, and practices.

**Keywords:** wellbeing economy; regenerative economy; sustainability scenarios; co-creation; eco-anxiety; degrowth

## 1. Introduction

The University of Denver's mission statement states that we are a 'great private university dedicated to the public good.' When this public good vision was formulated over twenty years ago, it generated specific university-wide scholarship, learning, and civic engagement goals. The scholarship goal is to "invigorate research and scholarship across the university to address important scientific, sociopolitical, and cultural questions of the new century". The learning goal is to "empower students to integrate and apply knowledge from across the disciplines and imagine new possibilities for themselves, their communities, and the world". The engagement goal is to "create active partnerships with local and global communities that can contribute to a sustainable society". In support of this mission, DU Provost Mary Clark recently launched an 'Ideas to Impact' project with the goal of identifying what could become transformational ideas for the University of Denver (DU). This campaign was designed to explore some of the most pressing challenges in our world and consider how DU faculty could help envision and co-create solutions with staff, students, and the public at large. Toward this end, there was a campus-wide call for 'big ideas', submitted in the form of white papers and vetted for potential support for teaching, research, and scholarship. One of the proposals selected as a major fundraising component is described here: The Center for a Regenerative Future (CRF).

Twenty thousand years ago, *Homo sapiens* represented less than 1% of the mammalian biomass on Earth. Today we represent 36% of Earth's mammalian biomass, and our livestock represents another 60%. Only 4% of Earth's mammalian biomass comprises wild

mammals [1]. This alarming expansion of humankind's impact on the biosphere is just one facet of what is described as the 'Great Acceleration' [2] of socio-economic activities that are driving climate change, ocean acidification, land degradation, resource depletion, life-altering pollution, and the resulting sixth mass extinction. These global trends have produced much human suffering and death and portend much more if left uninterrupted. We are rapidly accelerating into an increasingly unstable, unsustainable, and undesirable situation unless immediate steps are taken to envision a different future informed by our best understanding of biophysical reality.

Envisioning an alternative future that remediates environmental degradation and human suffering requires heeding the wisdom of the Baseball Hall of Famer and People's Philosopher Yogi Berra: 'If you don't know where you're going, you'll end up someplace else.' It requires identifying a broad suite of social, political, ecological, agricultural, industrial, educational, and urban practices that are regenerative. That is, identifying practices that not only can be perpetuated into the future (i.e., "sustainable") but that actually facilitate healing of the harm we have already inflicted. It also requires mustering the collective will to implement them. The Center for a Regenerative Future (CRF) will envision these futures and engage the public in ways that muster the collective will for their realization. The regenerative futures scholar Kimberly Camrass [3] emphasizes a definition of regenerative that is about rebirth, reformation, restoration, repairing of ecological damage, and, perhaps most importantly, "reclamation of greater social choice". Current structural power limits and narrows our social, political, and economic choices for moving forward [4].

The CRF leverages expertise in many areas of sustainability-related work by DU faculty in the sciences, social sciences, arts, and humanities. The CRF challenges structural power and presents 'choices' and 'alternatives' grounded in biophysical reality that are just, desirable, and not just sustainable but regenerative. There is growing recognition that contemporary 'Green Growth', 'Smart Growth', and other development paradigms do not acknowledge the urgent need to address current climate realities and trends [5] and are ill-equipped to accomplish this. Herein lies the intrinsic difference between sustainable and regenerative: A regenerative approach embraces the alternative, "degrowth" paradigm, which calls for planned reductions in energy and resource use that bring populations and economies back into balance with the living world in just and equitable ways [6]. It has been argued that degrowth will happen regardless of our actions and that the challenge is how much we acknowledge the biophysical reality of this challenge and take collective action to make the inevitable degrowth trajectory fairer and wiser.

## 2. Motivation for the Center

The CRF concept is responsive to student needs as well as a global moral imperative. Hickman et al. [7], in a survey of youth ages 16–25 across ten countries and four languages, found that nearly half of the youth surveyed ($n$ = 10,000) state that climate anxiety impacts their daily lives, 75% say that 'the future is frightening', 64% said that their governments were not doing enough to avoid climate catastrophe, and 39% are hesitant to have children. Here at DU, within the pre-existing Center for Sustainability, we had frequent discussions with students about the mental health ramifications of what they need to learn in courses ranging from biology to geography to social work about such issues as climate change and mass extinction. This content is not about the future but the current state of Earth due to anthropogenic harms. Their anxiety about the current and future ability of Earth to sustain them was impacting many facets of their lives and motivated our desire to create a CRF that better meets their needs.

Additionally, many young people are exploring alternatives to higher education, seeing the academy as antiquated and out of touch. Many perceive the debt incurred from attending as not worth their opportunity costs and financial loss [8]. They are urgently seeking mentors and role models willing and able to explore regenerative systems for working and living with them, which the CRF will support. The DU Center for

Sustainability has been the focal point for students in this arena by providing internship opportunities in a campus-based food bank, bike repair shop, and thrift store. However, we believe that the needs of students and the broader community were best served by re-fashioning the Center for Sustainability as the Center for a Regenerative Future, which not only provides the services of the original Center for Sustainability but would also be the umbrella under which academics, research, consulting, and DU operations would be coordinated. That is, becoming a true "center".

The CRF operates in the spirit of overcoming the mass production model of higher education that is increasingly being challenged by artificial intelligence applications such as ChatGPT. We envision using some potential of the internet in tandem with face-to-face teaching and workshops that integrate learning, critical thinking, research, policy, and outreach that are effective at bringing about systemic change. We hope to see the CRF evolve in a way that is consilient with the spirit of the meta-university described by Costanza and others [9].

The Provost's call for 'Big Ideas' resulted in eight white papers associated with the following topics: 'Climate and Society', 'Grand Challenges Health Equity', 'Center for Pathways to Access and Equity in Education', 'Institute for Housing Affordability and Strong Communities', 'Interdisciplinary Institute for Democracy', and 'Quantum at DU' in addition to the 'Center for a Regenerative Future.' Most, if not all, these proposals are essentially addressing a failure of governance with respect to our government's seeming inability to deal with addressing what has been traditionally recognized as market failures related to public goods and the provisioning of adequate social safety nets. Our vision for the CRF embraces all the 'big ideas' of these white papers from a holistic 'systems thinking' perspective. As stated by Costanza et al. [9]: "All of the earth's systems are interdependent, and real solutions to the current challenges must employ holistic, integrated analysis and creative, transdisciplinary education, and solutions. Higher education and academic research play a critical role in achieving sustainable well-being, not only in educating future leaders and producing knowledge but as an active agent in the co-production of real solutions".

Currently, the University of Denver has broad expertise in the areas of sustainability, conservation, and global environmental change. However, the spheres of instruction, research, DU operations, and consulting have not been well integrated, and many of us feel siloed within the university. Originally, our University's Sustainability Council was designed to connect individuals across campus by welcoming any student, faculty, or staff member to participate in fostering internal projects related to facilities management (e.g., recycling programs, tracking greenhouse gas emissions of the university), curriculum change (e.g., cataloging all courses across campus with sustainable content), and community engagement projects (e.g., the community garden, bio-blitz events). However, the Sustainability Council has been seen as lacking enough support or power to engage all appropriate partners and implement ideas that have a broader scope (e.g., expanding the campus community gardens to feed the unhoused or incorporating sustainability into all curricula). Despite the many past accomplishments of the Council (those listed above, plus sustainability conferences, installation of solar panels, LEED certification of new buildings, reductions in plastics, etc.), the COVID pandemic resulted in a resistance to unpaid labor and a healthy guarding of time that has crippled participation and effectiveness going forward. Furthermore, the Council has not operated as a place to facilitate interdisciplinary research. A different initiative at DU recently resulted in the bringing together of 40 faculty across campus who research aspects of global environmental change; those in different colleges became aware for the first time of significant overlap in interests. However, without ongoing support, those connections, confounded by the funding structure of the university, are likely to whither.

We envisioned the CRF as the foundation of a new organizational structure that reinvigorates the campus with a broader and more inclusive mandate and de-siloes existing efforts to enable all activities related to these fields to be connected. The CRF is a center of

research and innovation that facilitates transdisciplinary scholarship and identifies new funding sources that can support activities that impact the campus and far beyond.

## 3. Mission of the Center

The Center for a Regenerative Future sponsors and promotes a variety of activities, including basic and applied research, curriculum innovation, science communication, environmental media-making, and policy development for a re-imagined and regenerative future. The CRF serves the traditional purposes of higher education, often described as cognitive development, character development, social development, career development, and intergenerational transfer of knowledge. In addition to these traditional roles for higher education, we see a growing demand for the academy to help civilization chart a path to a regenerative and desirable future based on a well-being economy. The CRF is transformative in its approach and unique in the North American academy for its courage and capacity to face today's urgent challenges with honest realism and practical idealism. The CRF is:

–    Explicitly transdisciplinary and collectively owned.

The 'polycrisis' challenge we face is an 'all-hands-on-deck' situation that benefits from the collaborative contributions of people in the natural sciences, social sciences, arts, and humanities. Collective ownership is essential for a just, equitable, diverse, and inclusive (JEDI) center; as such, it is motivating for all participants to contribute to something greater than themselves and simultaneously obviating the turf wars that are so often characteristic of existing 'centers' in universities.

–    Cross-cultural and trans-historical in scope.

There is no 'one size fits all' solution to our societal challenges. There are many debates about the viability of mega-cities, green growth vs. degrowth, and alternative urban forms as they have manifested in the past. When multiple data sources and diverse interpretive perspectives are brought to bear on these problems, it increases the quality and resilience of the solutions that are generated. As to being 'trans-historical', we draw on the wisdom of Yogi Berra again: *'The only thing we learn from history is that we don't learn a damn thing from history'*. The CRF will do all it can to refute Yogi Berra's truth.

–    Utilize working groups as a vehicle for problem identification.

International and transdisciplinary working groups provide a vehicle for outreach to scholars, business leaders, NGOs, public advocates, and governmental officials to participate in helping each other understand problems from a variety of perspectives and experiences. This draws on the models of established synthesis centers (NCEAS, SESYNC) in which thought leaders address concerns we have heard from existing community partners. Many students, faculty, staff, and external university partners have argued for "paradigm-change" but are at a loss about how to accomplish this. This suggests that we need much more theoretical engagement with issues around sustainability and regeneration, much more basic research, and much better dissemination of scholarly work in modes that are accessible to relevant external communities and stakeholders. The CRF will incorporate all these functions of a Synthesis Center.

–    Be outward facing in terms of public outreach and media relations.

Public outreach and engagement is a fundamental pillar on which the CRF stands. Dialog with the public via public radio, social media, movies, print, and art exhibits enhances the diversity of our collaborators and dramatically extends the reach and exposure of the work we are doing. We partner with faculty in Media, Film, and Journalism Studies and faculty in Emergent Digital Practices to both promote our work and invite greater participation in CRF projects.

The CRF works with existing research and governance bodies at the University and re-arranges them within a new structure aimed at making the whole greater than the sum of its parts (Figure 1 presents one possible structure; others are imaginable). In many

respects, we are attempting to 'un-silo' individuals working in these areas who are unaware of other work taking place at the University. The project of transforming the Center for Sustainability into an expanded Center for a Regenerative Future honored the spirit of the university's legacy goals while updating them to meet the challenges of the modern age, including those related to global environmental change, massive human dislocations and migrations borne of climate change and economic deprivation, and rapidly growing cities teeming with un-housed individuals and entire communities. These convergent, "terricidal" trends (as described by Indigenous scholars and activists who are advancing much Regenerative Futures work) have produced a crisis of habitability that is unparalleled in human history [10,11].

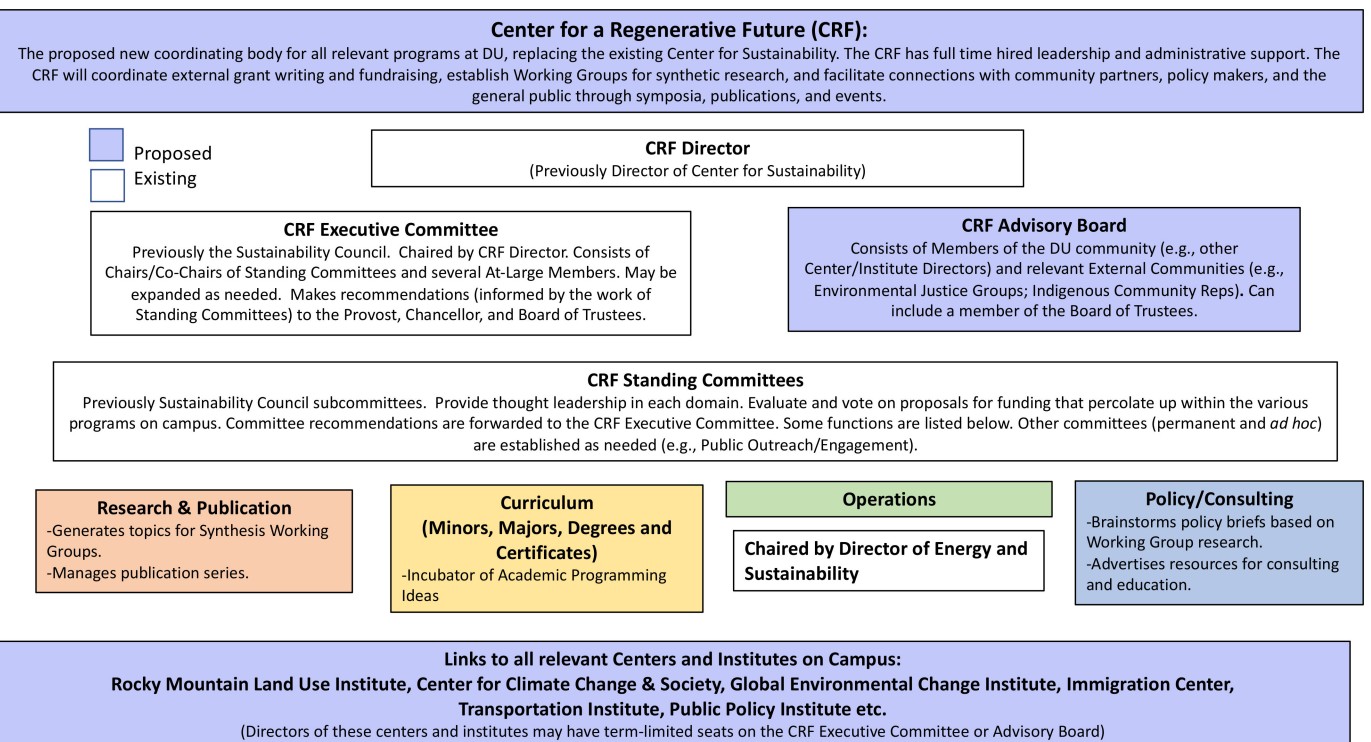

**Figure 1.** Proposed structure of CRF relative to existing and proposed units at the University of Denver.

The CRF furthers DUs commitment to the public good by targeting the polycrisis [12] with holistic, transdisciplinary approaches that forge a much closer connection between basic and applied research. It will serve as an incubator for curriculum and academic program development that engages with the alternative visions of habitability produced by decolonial and Indigenous critiques of traditional sustainability science [13]. It will explore, with students, new "imaginaries" for self and society and produce what Camrass [3] describes as "empowering narratives of hope and possibility". The CRF disseminates research results, innovative curriculum proposals, and policy briefs in modalities that can reach the broader public and agencies, including modalities that were unforeseen by the original author's vision of DUs Public Good.

The original RFP for the Ideas for Impact initiative asked, "How will this Big Idea change the world?" The CRFs plan for changing the world is a bit more modest: to help foster a *civilizational transition* that begins to heal our ecosystems, cities, and Earth at large. We harbor no fantasy that we can achieve this alone. One way we intend to start tipping the scale is by addressing students' need for guidance from responsible and caring mentors and role models who are on their side and believe they deserve a better future.

Jason Hickel, an anthropologist, and author of *Less is More: How Degrowth Will Save the World,* says that degrowth is, ultimately, a process of *decolonization*: of lands, peoples, and

minds [14]. He says it stands for "de-enclosure of the commons, de-commodification of public goods, de-intensification of work and life, de-thingification of humans and nature, and de-escalation of ecological crisis". We might add to this list the deconstruction of traditional ways of academic being and belonging. This is a holistic and powerful view. It is a view that has clear implications for the university's fulfillment of JEDI commitments. It charts a path forward that breaks from business-as-usual on the DU campus.

When a question was posed to a finalist in the university's recent (and unsuccessful) search for a chief Sustainability Officer about what sustainability would look like at DU, the answer regarded eliminating plastic cups, lowering classroom temperatures, and taking other measures to reduce our carbon footprint. Although difference-making, this is instrumentalist thinking in service of a traditional sustainability agenda. Un-mentioned were much bigger issues such as eliminating administrative bloat, re-structuring academic life and policy to better facilitate interdisciplinary teaching and research, promoting workload equity (an aspect of "de-intensifying" faculty life), evolving curriculum to accommodate Indigenous ways of knowing (e.g., moving from "Environmental Science" to "Environment and Land Studies"), investing in student explorations of deceleration of the human footprint, and embracing of true ecological economics within human carrying capacity. We cannot, as a campus community, be afraid to go in any of these directions. The CRF operates with them in mind.

Scholars have demonstrated the importance of cultural diversity to the long-term resilience of human societies [15,16]. Compositional diversity is emerging as key to the long-term survival of American universities. Increasing compositional diversity at DU is a very tough sell given our settler-colonial history, cost of attendance, and reputation. Our founders are unequivocally connected with the Sand Creek Massacre, the slaughtering of an estimated 150 Cheyenne and Arapaho children, women, and men by U.S. military forces [17]. Their ghosts implore us to achieve what we can to heal our past harm and elevate the voices of those who have been silenced. We believe our history gives us a particular responsibility in this regard.

The CRF hosts working groups on topics that matter to and/or intersect with the existential concerns of marginalized communities, e.g., environmentally-just urban planning, Indigenous Knowledge for climate change adaptation, biodiversity regeneration and healing, food sovereignty, etc. We believe that if we make serious efforts to decolonize the university in the ways described by Hickel [14] and make concerted efforts to actively promote what the CRF stands for, that compositional diversity will follow.

More tractable than the challenge of increasing compositional diversity is the challenge of increasing the inclusion of faculty, students, and staff in Regenerative Future work. The Center's name signals accountability to all people, the diversities of life on Earth, and future generations of all species, including our own. It signals a serious interest in engaging with Traditional Ecological Knowledge and Indigenous wisdom, theory, and practice. Having the CRF centrally housed and not within the domain of a particular academic unit or under the control of a particular faculty is a significant decolonizing move. It indicates a serious interest in fostering collective ownership, stewardship, and guardianship of the Center's activities. We are committed to exploring new structural arrangements for governing the center that would better integrate into its operation existing entities such as the Sustainability Council and Facilities Management.

It has been said by scholars doing environmental justice work that our greatest challenges are not simply scientific and technological [18]. Rather, they go much deeper: they are spiritual (in E.O. Wilson's sense of the sanctity of nature) and cultural. Too many efforts to meet JEDI objectives, no matter how well-meaning, act as cover for continuing Anglo-European coloniality. To increase compositional diversity at DU, we need to create *social infrastructures* (physical spaces and campus climates) that are welcoming, inclusive of diverse world views and research/teaching agendas (what decolonial scholars sometimes refer to as *pluriverses*), and truly participatory. The CRF provides one such space.

## 4. Institutional Challenges

Upton Sinclair's famous quote, "*It is difficult to get a man to understand something, when his salary depends on his not understanding it*". is relevant to the challenges we experienced in presenting our idea for a Center for a Regenerative Future. In a similar and analogous vein, it is difficult to solicit funding for programs aimed to bring about 'systemic change' from wealthy individuals and entities that have benefitted dramatically from the very system you hope to change. To wit, we have been strongly advised by many people in our administration to avoid the very phrase 'Degrowth' when describing the CRF because it is unlikely to be supported by a system that depends on growth. Bill Gates, a noted billionaire philanthropist, stated: "*I don't think it's realistic to say that people are utterly going to change their lifestyle because of concerns about climate*". Bill Gates is one of many techno-optimists that believe we can innovate our way out of the polycrisis. Other techno-fixologists such as Mark Jacobsen (electrify everything) and David Keith (geo-engineering and carbon capture) propose technological fixes to the climate crisis that are representative of the 'Green Growth' camp [19,20]. Jacobsen and Keith have very different ideas as to how technological adaptations to the climate crisis will manifest; nonetheless, they both offer hope and counterarguments for those afraid of degrowth and the systemic changes it demands. An interesting and compelling case against both Jacobsen and Keith's technological solutions is discussed on this Crazy Town podcast.

The EU Parliament seems to be divided on the question of degrowth, but there is a growing trend toward embracing degrowth. Twenty European members of Parliament wrote an open letter at a 2023 Beyond Growth conference in Brussels that calls for the following: (1) Post-Growth European Institutions, (2) European Green Deal, (3) Beyond growth policies based on the four principles of Biocapacity, Fairness, Wellbeing, and Democracy. The letter states that there is no empirical basis suggesting that decoupling economic growth and environmental decline is possible, degrowth is a peace project, smaller ecological footprints do not mean lower levels of well-being, and market forces exacerbate rather than alleviate our most serious problems, particularly when paired with extreme levels of economic inequality. Nonetheless, taking the case for degrowth seriously–as well as the case for decolonizing structures and minds–is an uphill battle in which the CRF will be engaging. It is both a moral imperative and an institutional challenge.

Low levels of ecological literacy and shifting baselines of environmental conditions are also a challenge. There is growing evidence that ecological literacy will be essential to developing sustainable human settlements [21]. The CRF will embrace the idea that biophysical limits and the laws of thermodynamics are not negotiable. As urbanization continues apace, our levels of ecological literacy are consequently degraded because of reduced exposure to natural environments [22]. In a discussion amongst our own faculty, we received aggressive questions such as: '*How can we have exceeded our carrying capacity if we are still growing*?' Questions such as this seem to challenge the very premise of a 'polycrisis' associated with the great acceleration. The many 'World Scientists' Warnings to Humanity' papers, in addition to the IPCC and IPBES reports and Millennium Ecosystem Assessments, be damned and ridiculed. Studies have shown that people trained in finance and economics have lower levels of ecological literacy [23]. These people often exist in the highest levels of business and government and control the allocation of many financial and non-financial resources. Shifting baseline syndrome (SBS) is a documented phenomenon where gradual changes in the accepted norms for the state of the natural environment manifest because people do not know what it used to be like [24]. This is a more sophisticated idea that is often summarized with the apocryphal and likely not true story of the 'Frog in the boiling pot of water.' It is very difficult to make people appreciate the consequences and implications of environmental change that they are not experiencing directly.

## 5. Discussion

Humankind is currently facing a civilizational crisis of multiple proportions. This polycrisis includes a failing economic system, collapsing ecological systems, and a rapidly

evolving technological system that is outpacing social, cultural, and biological evolution. Worldviews also experience crises. Copernicus and Einstein were 'radical' thinkers who shattered the worldviews of astronomy and physics, yet the disciplines of Astronomy and Physics survived and evolved with the benefit of these significant disruptions to the status quo. Today, the academic discipline of Economics is in crisis; traditional economic thinking founded in a growth imperative has dire impacts on real-world policy and practice. "Sustainability" in the traditional sense is now making way for the more innovative reality of regenerative futures. The Royal Society of Arts defines regenerative futures as a way "*we can re-think our relationship with the planet and allow healthy ecosystems, as an interconnecting whole across environment, society and economy*". In many respects, this embraces an ecological economics paradigm in which human, social, and built capital are enveloped by and exist within the natural capital provided by our planet (Costanza et al. [25]. The interaction of Natural, Social, Human, and Built capital is the source of human well-being (Figure 2). The new paradigm calls for bringing populations and economies into balance with the living world in just, equitable, and regenerative ways.

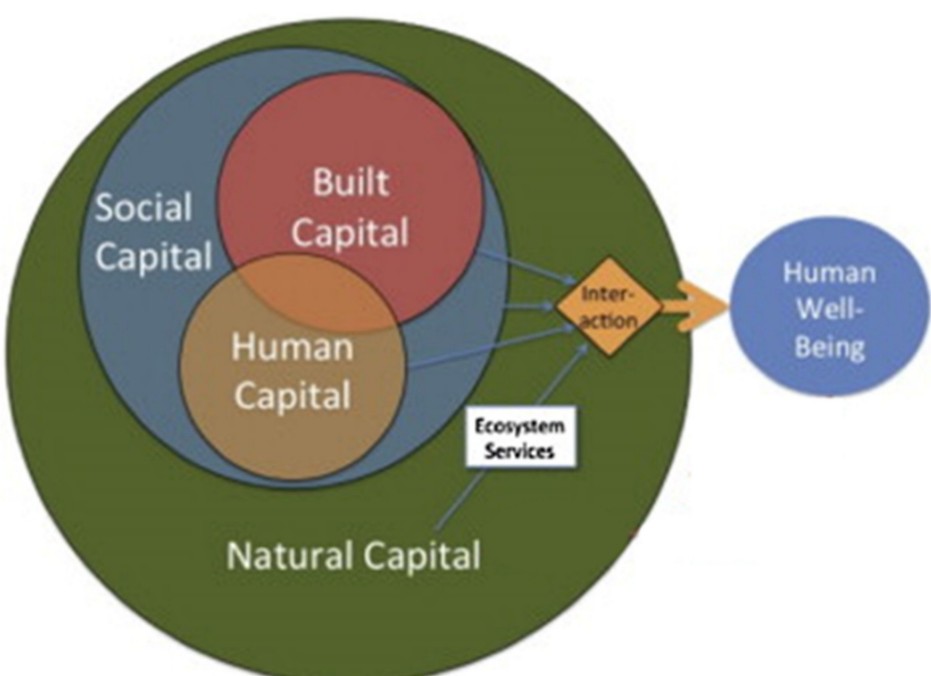

**Figure 2.** Interaction of Natural, Social, Human, and Built Capital producing human well-being. (Figure Adapted from Costanza et al. [25].

## 6. Conclusions

This paper describes our effort to reinvent an existing Center for Sustainability as a Center for a Regenerative Future. This reinvention involved incorporating elements of what has proven to be a successful "Synthesis Center" research model into Center functions and activities. The CRF promotes regenerative futures theory, design, and practice by utilizing concepts such as systems thinking, mutual aid, and traditional ecological knowledge. It will sponsor an array of international research collaborations, interdisciplinary curriculum-building initiatives, and public outreach events and activities. We propose a publication series for policy briefs and dissemination of expert knowledge in other modalities, including traditional peer-reviewed scholarship and public outreach via journalism and the visual and performing arts. We do not know whether our reinvention will be successful, given competing perspectives on important issues, internal fears of structural change, and the wishes of external donors. However, what is abundantly clear is that business as usual has brought us to the eve of destruction. Disruption to the status quo, in some form, is necessary.

**Author Contributions:** S.M.B.: Consptualization, Project Administration, Resources. D.S.: Consptualization, Investigation, Project Administration, Resources, Validation. A.S.: Consptualization, Methodology, Resources. P.S.: Consptualization, Investigation, Project Administration, Resources. All authors have read and agreed to the published version of the manuscript.

**Funding:** This research received no external funding.

**Institutional Review Board Statement:** Not applicable.

**Informed Consent Statement:** Not applicable.

**Data Availability Statement:** No new data were created.

**Conflicts of Interest:** The authors declare no conflict of interest.

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
