# Peer review of "Co-Creation of a Center for a Regenerative Future"

_sustainability, doi:10.3390/su151712861_

Round 1

Reviewer 1 Report (Previous Reviewer 1)

Accept in the current

Author Response

Thank you for your time and consideration. In response to the reviewers comments we have created a much shorter conclusion and added a discussion section that addresses concerns about citations in the conclusion and too long a conclusion. As to first person writing we have a mixed approach which is recognized as appropriate in some circumstances. We avoid making first person opinion statements.

Reviewer 2 Report (Previous Reviewer 2)

It is written well, however, it would be better if authors could broaden the idea for the understanding of the international audience of the journal.

The citation should be in journal's format. The citation is missing from some paragraphs.

Concise the conclusion part, some part needs to be moved to the discussion part. There should not be any citation in the conclusion.

its okay, minor improvements.

Author Response

Thank you for your time and consideration. In response to the reviewers comments we have created a much shorter conclusion and added a discussion section that addresses concerns about citations in the conclusion and too long a conclusion. As to first person writing we have a mixed approach which is recognized as appropriate in some circumstances. We avoid making first person opinion statements.

Reviewer 3 Report (Previous Reviewer 3)

Write in the third person. One should not personalize knowledge. For example, in the abstract remove the terms "we" and "our"...

I agree, given the nature of the manuscript, that it should be submitted as a project report.

Author Response

Thank you for your time and consideration. In response to the reviewers comments we have created a much shorter conclusion and added a discussion section that addresses concerns about citations in the conclusion and too long a conclusion. As to first person writing we have a mixed approach which is recognized as appropriate in some circumstances. We avoid making first person opinion statements.

This manuscript is a resubmission of an earlier submission. The following is a list of the peer review reports and author responses from that submission.

Round 1

Reviewer 1 Report

I have gone through the manuscript, I find it difficult to make a conclusion as the manuscript has not presented any empirical data or quantitative data for argument. On this basis, I do not recommend the manuscript for publication. I think the manuscript may be suited for another journal.

Reviewer 2 Report

Comments to the authors:

The present study is about “the ideas, conditions, and environments that motivated co-creation of a Center for a Regenerative Future at the University of Denver”. The authors have tried to narrate the scope and significance of the Center for a Regenerative Future at the University of Denver but totally failed to justify the core scope and objectives of the center for narrating the story.

 The aims and objectives of the paper are not clear. Make it sure that the objectives of the paper are clear, so I suggest you form 3 objectives of the study, then center your narration around it. I do not understand what kind of paper it is? Original article or review article or perspective article. The paper needs to be broken down into introduction, study area, material and methods, results and discussion, conclusion. The authors just have a narration on the mission of the center. The story authors are going to narrate is not clear and focused. It should be crystal clear to understand what the authors are trying to say? Sometimes is around the Center for a Regenerative Future and suddenly moving the EU parliament, why? I recommend you revised the Ms thoroughly.

Specific comments:

Author's affiliation is not correct, which state and country?

Abstract- This is part needed to be re-written after a thorough revision of the Ms.

Background: Does background is needed? I don't think so, it is needed.

Introduction- Line-98-109 No citation. The role and scope of the Center for a Regenerative Future is need to be highlighted in the text.

 Line 160; could you please briefly tell what kind of basic and applied research?

Figure 2 is just copied from the paper, if you have adapted it from the paper, at least authors need some modification, how can you copy and paste the figure as such, are you familiars with the ethics/copy right in publication.

References: Format the reference according to the journal’s requirements. Very less references?

Editorial is needed for the Ms

Reviewer 3 Report

Major Comments
The authors effectively address a topic of great interest in their study.
They extensively utilize relevant and up-to-date literature.

Suggested Revisions
The authors' study has noteworthy practical and social implications.  It should also point out the contributions to the literature.

Reviewer 4 Report

I have received the article entitled "Co-Creation of a Center for a Regenerative Future" for review for the Sustainability journal and I am providing my comments below.

The topic discussed in the article, which covers sustainable civilizational development, degrowth, and climate change, is of utmost importance. The article presents an intriguing proposition: establishing a Center for a Regenerative Future at the University of Denver to coordinate research, education, and promotion of sustainability. The authors envision this regenerative future as one that not only maintains socio-economic and environmental balance but also provides healing for the damage inflicted by rapid civilizational growth.

In general, the subject of sustainability and the idea of a regenerative future make the article highly engaging and valuable. The case study of the proposed Center for a Regenerative Future at the University of Denver is particularly noteworthy.

However, during my review, I encountered several significant issues that need addressing prior to potential publication in a scientific journal. Here is a list of these concerns:

  • The '0. Background' section should be merged with the
    '1. Introduction'. The background information related to the establishment of the Center for a Regenerative Future can be incorporated into the Introduction without necessitating a separate section.
  • The Introduction requires considerable enhancement. It lacks a clear statement of the research question, aims, or scientific premises. Given that the text is an article, it must explicitly state the research aims or questions and the novelty that the article brings to the field.
  • I have a general concern regarding the Center for a Regenerative Future at the University of Denver. Based on the information provided in the text, it appears to be a project for reinventing an existing Center for Sustainability that has not yet been implemented. Therefore,
    a question of the appropriateness of a yet-to-be-realized project as the basis for a research article remains open.
  • The article does not provide information about the research methodology used. If any research methods were applied (qualitative or quantitative), these should be clearly described, as it's vital for a scientific article to include a section on research methods.
  • While there are numerous current references cited on topics such as degrowth, climate crisis, and a regenerative future, there appears to be a lack of systematic literature review. This should be conducted and provided in a separate section that clearly ties the topic to the existing body of knowledge.

In conclusion, while the topic is crucial and the text engaging, substantial improvements are necessary. The introduction should include clearly stated research questions and aims. A systematic literature review should be undertaken, covering the transition from sustainable development to a regenerative future and linking the text to the existing knowledge base. The fact that the article discusses a planned project rather than an established venture should be stated explicitly in the introduction rather than in the conclusions. Furthermore, the methodology used should be clearly outlined, and the initial two sections should be merged. After these changes, the article would make a valuable and intriguing contribution that could be reconsidered for publication in a scientific journal.